Accepted at the ICLR 2024 Workshop on AI4Differential Equations In Science

# GA-ReLU: AN ACTIVATION FUNCTION FOR GEOMETRIC ALGEBRA NETWORKS APPLIED TO 2D NAVIER-STOKES PDEs

**Alberto Pepe, & Joan Lasenby**
Signal Processing and Communications Lab
University of Cambridge
Cambridge, UK
{ap2219,jl221}@cam.ac.uk

**Sven Buchholz**
Department of Computer Science and Media
Technical University Brandenburg
Brangenburg, Germany
sven.buchholz@th-brandenburg.de

## ABSTRACT

Many differential equations describing physical phenomena are intrinsically geometric in nature. It has been demonstrated how this geometric structure of data can be captured effectively through networks sitting in Geometric Algebra (GA) that work with multivectors, making them suitable candidates to solve differential equations. GA networks however, are still mostly uncharted territory. In this paper we focus on non-linearities, since applying them to multivectors is not a trivial task: they are generally applied in a point-wise fashion over each real-valued component of a multivector. This approach discards interactions between different elements of the multivector input and compromises the geometric nature of GA networks. To bridge this gap, we propose GA-ReLU, a GA approach to the rectified linear unit (ReLU), and show how it can improve the solution of Navier-Stokes PDEs.

## 1 INTRODUCTION

Geometric (or Clifford) Algebra (GA) has been recently rediscovered as a suitable mathematical space in which to build neural networks that are *truly* geometrical (Brandstetter et al., 2022; Ruhe et al., 2023). While not new (Pearson & Bisset, 1994), GA networks have only recently been employed successfully in computational biology (Pepe et al., 2024a), physics (Ruhe et al., 2023), vision (Pepe et al., 2024b) and partial differential equations (PDEs) (Brandstetter et al., 2022). Data in all these fields presents an intrinsic geometric structure that can be captured by operating on multivectors. In higher dimensional Clifford Algebras, however, it is not easy to define the concept of differentiability and hence construct an expressive function theory. Ever since the early proposals of Clifford neurons in Pearson & Bisset (1994); Buchholz & Sommer (2001); Arena et al. (1994) until today, activations have always been applied onto multivectors element-wise. When doing so, we are losing part of the geometric coupling of data that GA networks strive to achieve: is there a better way to apply non-linearities to multivectors? We try to address this issue in this paper.

## 2 PROBLEM DEFINITION

We wish to define a non-linear function $\psi(\mathbf{x}) : \mathcal{G}_n \to \mathcal{G}_n$ for networks operating in GA in order to extend the geometric flavour of the approach also to the activation function. In particular, we refer to the Navier-Stokes PDE problem as formulated in Brandstetter et al. (2022) and train two networks, namely the Clifford ResNet and Clifford Fourier Neural Operator (FNO) with ReLU and GA-ReLU activation functions, respectively.

The incompressible Navier-Stokes equations in 2D are given by:

$$\frac{\partial \mathbf{u}}{\partial t} + (\mathbf{u} \cdot \nabla)\mathbf{u} = -\frac{1}{\rho}\nabla p + \nu\nabla^2\mathbf{u} + \mathbf{f}, \quad \nabla \cdot \mathbf{u} = 0, \tag{1}$$

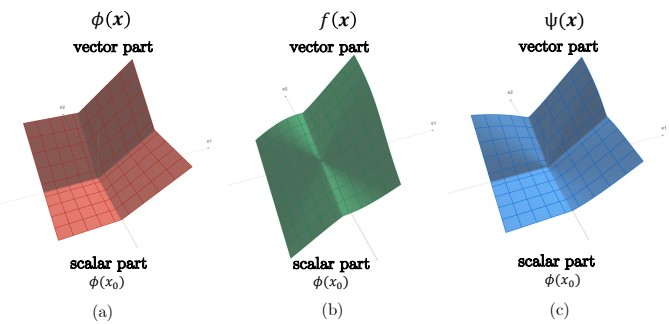

$\phi(\boldsymbol{x})$     $f(\boldsymbol{x})$     $\psi(\boldsymbol{x})$

vector part     vector part     vector part

scalar part     scalar part     scalar part

$\phi(x_0)$     $\phi(x_0)$     $\phi(x_0)$

(a)     (b)     (c)

Figure 1: Vector and scalar part of the (a) coefficient-wise ReLU $\phi(\mathbf{x})$ (b) phase-dependent ReLU $f(\mathbf{x})$ and (c) GA-ReLU $\psi(\mathbf{x})$

in which $\frac{\partial \mathbf{u}}{\partial t}$ is the time derivative of the fluid velocity vector $\mathbf{u} = (u_x, u_y)$, $(\mathbf{u} \cdot \nabla)\mathbf{u}$ is the convective term, $-\frac{1}{\rho}\nabla p$ is the pressure gradient, $\nu \nabla^2 \mathbf{u}$ is the viscous diffusion and $\mathbf{f}$ is the external force term. The incompressibility of the fluid is ensured by $\nabla \cdot \mathbf{u} = 0$.

Since there exists a coupling between vector quantities (the velocity $\mathbf{u}$) and scalar ones (the pressure field $p$, or the smoke $s$ advected by $\mathbf{u}$), it makes sense to express this coupling by "wrapping" together the vector and scalar information as a single multivector of the type:

$$\mathbf{x} = s \cdot 1 + u_{e_1} e_1 + u_{e_2} e_2, \tag{2}$$

where $u_{e_1}$ and $u_{e_2}$ are the components of $\mathbf{u}$ along the $e_1$ and $e_2$ direction, respectively. Given two multivectors $\{\mathbf{x}_{t_i}, \mathbf{x}_{t_{i+1}}\}$ at two different time instants $t_i, t_{i+1}$, we want to use a machine learning pipeline that estimates $\mathbf{x}_{t_{i+2}}$. This multivector approach has been demonstrated to be more successful than estimating $(s, u_x, u_y)_{t+2}$ independently through a network not in GA, since the coupling between different geometric quantities has to be inferred by the network rather being explicitly expressed through a multivector structure. We wish to keep the same coupling also when applying non-linearities.

## 3   GA-RELU: A RELU IN GEOMETRIC ALGEBRA

We want to design $\psi(\mathbf{x})$ in such a way that (i) it preserves the behaviour of the equivalent activation function defined over $\mathbb{R}$, and (ii) it is able to differentiate between grades and model interactions between components of the same grade.
An activation function $\phi$ (that we will assume from now on to be the ReLU function) has commonly been applied to a multivector $\mathbf{x}$ element-wise, i.e.:

$$\phi(\mathbf{x}) = \sum_{i=0}^{2^n} \phi(\mathbf{x}_i), \tag{3}$$

where $\mathbf{x}_i$ is the $i$-th blade component of $\mathbf{x}$ and $n$ is the space dimensionality. This approach satisfies (i), but not (ii) (see Fig.1a). To fill in this gap, we introduce GA-ReLU. GA-ReLU is the composition of (1) a coefficient-wise ReLU, that depends on the magnitude of each multivector coefficient, and (2) a phase-dependent ReLU, that grasps the interaction between vector coefficients and depends on their phase difference. We will derive GA-ReLU below.

We look at the complex domain and express our 2D multivector in terms of complex numbers. The phase-dependent ReLU has been inspired from the complex cardioid activation function firstly introduced in Virtue et al. (2017), defined as

$$f(z) = \frac{1}{2}\left(1 + \cos(\arg(z))\right) z = K(\arg(z))z, \tag{4}$$

in which $z = a + bi \in \mathbb{C}$, $i^2 = -1$ and $K$ an attenuation function dependent on the argument of $z$. The complex cardioid is an extension of the ReLU function over $\mathbb{C}$ and it is dependent only on the phase of the input rather than on its magnitude.

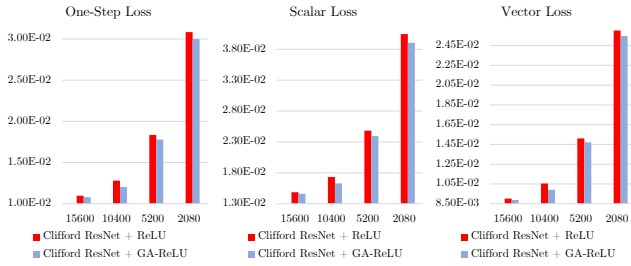

Figure 2: Errors versus number of training data for Clifford ResNet with ReLU and GA-ReLU activation functions.

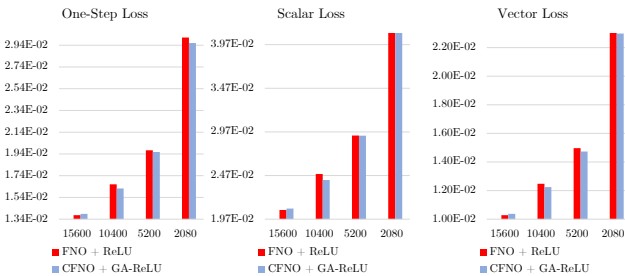

Figure 3: Errors versus number of training data for Clifford FNO with ReLU and GA-ReLU activation functions.

It is known that the 2D Geometric Algebra $\mathcal{G}_{2,0}$ is isomorphic to $\mathbb{C}$ by simply taking $I \triangleq e_{12}$ as our imaginary unit, since $e_{12}^2 = (e_1 e_2)^2 = e_1 e_2 e_1 e_2 = -e_1 e_1 e_2 e_2 = -1 = i^2$. By keeping this in mind, it is easy to see that a generic multivector $\mathbf{x} = x_0 + x_1 e_1 + x_2 e_2 + x_{12} e_{12} \in \mathcal{G}_{2,0}$ can be decomposed into "a sum of two complex numbers" as follows:

$$\mathbf{x} = (x_0 + Ix_{12}) + e_1(x_1 + Ix_2) = z_S + e_1 z_V. \tag{5}$$

Following Brandstetter et al. (2022), we will refer to $z_S$ as the spinor part and to $z_V$ as the vector part. We can then evaluate Eq.4 on $\mathbf{x}$ as follows:

$$f(\mathbf{x}) = f(z_S) + e_1 f(z_V). \tag{6}$$

In 2D Navier-Stokes there is no bivector component, so Eq. 6 reduces to:

$$f(\mathbf{x}) = K(\arg(z_S))z_S + e_1 K(\arg(z_V))z_V = \phi(x_0) + e_1 K(\arg(z_V))z_V \tag{7}$$

since $K(\arg(z_S)) = K(\arg(0)) = 1$ for $x_0 > 0$ and $K(\arg(0)) = 0$ for $x_0 \leq 0$. On the other hand, the second term can be computed to be:

$$e_1 K(\arg(z_V))z_V = K(\tan^{-1}(\tfrac{x_2}{x_1}))x_1 e_1 + K(\tan^{-1}(\tfrac{x_2}{x_1}))x_2 e_2. \tag{8}$$

Eventually, we obtain that

$$f(\mathbf{x}) = \phi(x_0) + K(\arg(z_V))x_1 e_1 + K(\arg(z_V))x_2 e_2, \tag{9}$$

meaning that the complex ReLU acts like a real ReLU over the scalar part $x_0$ and attenuates the vector components $x_1, x_2$ by an amount proportional to the phase between them. Eq.9, however, is unbounded for negative vector components (see Fig.1b), which could cause numerical instability and negatively impact convergence. Hence, we would still want to keep the element-wise ReLU on the vector components to guarantee also a dependence on their magnitude. The final expression of GA-ReLU will then be:

$$\psi(\mathbf{x}) = (\phi \circ f)(\mathbf{x}) = \phi(x_0) + \phi(K(\tan^{-1}(\tfrac{x_2}{x_1}))x_1)e_1 + \phi(K(\tan^{-1}(\tfrac{x_2}{x_1}))x_2)e_2. \tag{10}$$

In Eq.10 we have the advantage of having a magnitude scaling similar to a ReLU (i.e. 0 output for negative input), but also a phase dependency deriving from the complex ReLU (see Fig.1c).

## 4 EXPERIMENTS

We generated our own dataset of fluid in motion over a regular grid through PhiFlow (Holl et al., 2020) following the specifications of Brandstetter et al. (2022) (see Appendix A.1). It is composed of 15600, 4680 and 3120 training, validation and testing sequences, respectively. We call "sequence" a pair of inputs and targets $(\mathbf{x}_{t_i}, \mathbf{x}_{t_{i+1}}; \mathbf{x}_{t_{i+2}})$. We trained a Clifford ResNet and a Clifford FNO with standard ReLU and GA-ReLU activation functions (see Appendix A.2). We report the summed mean squared error over the three multivector coefficients (One Step Loss) and over the scalar (Scalar Loss) and vector (Vector Loss) coefficients.

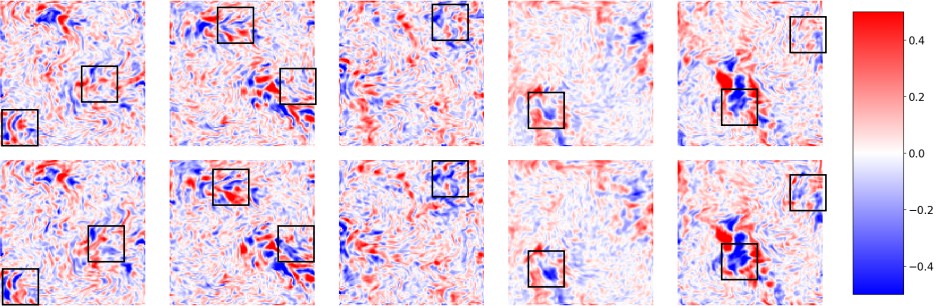

Figure 4: Difference between ground truth and predicted scalar fields $s_{t_{i+2}} - \hat{s}_{t_{i+2}}$ for 5 different time instants. Top row: GA-ReLU, bottom row: ReLU. Higher intensity is worse.

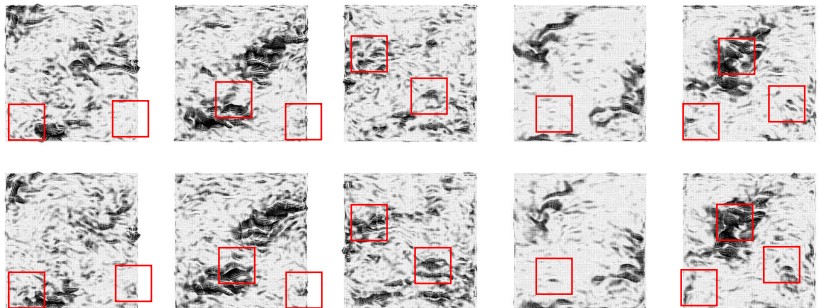

Figure 5: Difference between ground truth and predicted vector fields $\mathbf{u}_{t_{i+2}} - \hat{\mathbf{u}}_{t_{i+2}}$ for 5 different time instants. Top row: GA-ReLU, bottom row: ReLU. Higher intensity is worse.

## 5 RESULTS

The 3 metrics measured for datasets of different sizes are reported in Fig.2 and in Fig. 3 for the Clifford ResNet and Clifford FNO, respectively. Note how, albeit small, the improvement from GA-ReLU is consistent for different dataset sizes. We plot the difference between ground truth and predicted scalar fields $s, \hat{s}$ for 5 different sequences in the test set in Fig. 4 and the difference between ground truth and predicted vector fields $\mathbf{u}, \hat{\mathbf{u}}$ in Fig.5. Note how, despite having minimally modified the activation function, it is possible to identify for each frame regions that deviate more from ground truth (i.e. areas in which the PDE solution is less exact) when employing an activation function that treats multivector components independently.

## 6 CONCLUSIONS

We introduced GA-ReLU, a modified version of ReLU for multivector-valued networks that attempts to take into account the coupling between multivector coefficients, and showed how it can improve over the baseline error for a 2D Navier-Stokes PDEs problem. GA-ReLU has the limitations

of being designed as an adaptation real-valued activation (ReLU) and for a specific mathematical space ($\mathcal{G}_{2,0}$). Nevertheless, we hope that GA-ReLU can highlight the importance of non-linearities that take into account the structure of multivectors.

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

## A   APPENDIX

### A.1   DATA GENERATION

We evaluate Navier-Stokes PDEs over a regular square grid of size $128 \times 128$ with resolution $\Delta x = \Delta y = 0.25$. The fluid has a viscosity of $\nu = 0.01$ and a buoyancy factor of $0.05$. The scalar field $s$ at $t = 0$ is intialised with Gaussian noise over a centered grid, while the vector field $\mathbf{u}$ at $t = 0$ is initialised to be 0 throughout over a staggered grid. We run the fluid dynamics simulations via PhiFlow for $21s$ and sample every $\Delta t = 1.5s$. We start collecting data after $\tau_0 = 4s$ to move away from initial conditions.

### A.2   TRAINING DETAILS

The Clifford ResNet in $\mathcal{G}_{2,0}$ has 4 residual blocks with 2 Clifford convolutional layers each, kernel size $3 \times 3$ and 64 hidden channels for a total of 2.4M trainable parameters.
The Clifford FNO in $\mathcal{G}_{2,0}$ has 4 FNO blocks, 6 Fourier modes for the $x, y$ components and 48 hidden channels, for a total of 38M trainable parameters. We trained the Clifford ResNet on data with batch size of $B = 16$ and the Clifford FNO on data with batch size of $B = 32$. Both networks have been trained for at most 200 epochs, implementing early stopping monitoring validation loss with patience $P = 15$ for the Clifford ResNet and $P = 10$ for the Clifford FNO. We minimized the One Step loss between ground truth $\mathbf{x}_{t+2}$ and predicted $\hat{\mathbf{x}}_{t+2}$ using Adam optimizer and fixed learning rate of $\eta = 10^{-4}$. Results produced have been averaged over 3 random seeds. Training has been performed on a single GPU NVIDIA GeForce RTX 4090, taking $64s$ per batch to train the Clifford

FNO and $48s$ per batch to train the Clifford ResNet. As expected, the activation function does not impact training time. Code can be found here.

