# OpenReview forum: "GA-ReLU: an activation function for Geometric Algebra Networks applied to 2D Navier-Stokes PDEs"
_ICLR.cc/2024/Workshop/AI4DiffEqtnsInSci — AI4DiffEqtnsInSci @ ICLR 2024 Poster_

### Official Review · Reviewer_HHf2 · 2024-02-22
**Good Paper, Accept**

**Rating:** 7
**Confidence:** 4

**Review:**

Congratulations to the authors in getting this research to this stage. I enjoyed reading this nicely drafted paper. Draft starts with a short, concise and excellent Introduction and defines the problem very well.

The title and abstract focus on the proposal of a new activation function for geometric algebra however I fail to grasp the novelty in the paper. The approach of combining two existing components to formulate the GA-RELU activation function and its subsequent application to a new domain is both commendable and indicative of the paper’s creative efforts to advance the field. The application is definitely new and hence makes the case for this paper. However, upon careful review,  the novelty of the concept of GA-RELU prompts further questions. The description and promotion of complex cardioid  activation under a new name of GA-RELU may overstate its uniqueness.

- Quality
  - The written quality of paper is good. Draft nicely explains the problem and solution. The paper use the modified complex cardioid  activation for the GA and demonstrate the improvement with this approach and
  - Draft clearly associates the physical meaning associated with mathematical equations used and derived.

- Clarity
  - The text is clear and describe the problem and solution with all necessary mathematical details.

- Originality
  - The concept of modifying the complex cardioid  activation is original and has not be proposed in the filed earlier.
  - Moreover, the application of GA-RELU for Geometric Algebra Networks is new and its application to the 2D NS PDE.

- Significance
  - The proposed research mark an significant improvement for Geometric Algebra Networks, however the experimentation uses on one example.
  - Due to limited experimental results it is hard to judge its usability and applications in different problems in GA.

---

### Meta-Review · Area_Chair_xQrq · 2024-03-03

**Recommendation:** Accept (Poster)

**Metareview:**

The reviewer marks this paper as a clear accept and I agree. There are some questions of novelty that I encourage the authors to address in the camera-ready version.

---

### Decision · Program_Chairs · 2024-03-03

Accept (Poster)